Support for the intermittent upwelling hypothesis using 10 years of barnacle recruitment data from a western ocean boundary in Atlantic Canada

Scrosati Ricardo A. rscrosat@stfx.ca 1
Ellrich Julius A. 2
1 Department of Biology, St. Francis Xavier University , Antigonish , Nova Scotia , Canada
2 Biologische Anstalt Helgoland, Shelf Sea System Ecology, Alfred Wegener Institute , Helgoland , Germany
Jeffery Nicholas
Electronic publication date: 2025 May 19
Publication date: 2025
Volume: 13
Electronic Location ID: e19470
Received 2025 Jan 31; Accepted 2025 Apr 23
Copyright: ©2025 Scrosati and Ellrich
Copyright year: 2025
Copyright holder: Scrosati and Ellrich
License: This is an open access article distributed under the terms of the Creative Commons Attribution License, which permits unrestricted use, distribution, reproduction and adaptation in any medium and for any purpose provided that it is properly attributed. For attribution, the original author(s), title, publication source (PeerJ) and either DOI or URL of the article must be cited.
License URL: https://creativecommons.org/licenses/by/4.0/

Keywords: Intertidal, Barnacle, Recruitment

Funding: Discovery Grant 311624 Natural Sciences and Engineering Research Council of Canada (NSERC) This research was funded by a Discovery Grant (number 311624) awarded to Ricardo A. Scrosati by the Natural Sciences and Engineering Research Council of Canada (NSERC). The funders had no role in study design, data collection and analysis, decision to publish, or preparation of the manuscript.

==============================
Recruitment is a key demographic step for the persistence of populations, so understanding its drivers has traditionally been a relevant goal of ecology. On marine rocky shores, coastal oceanography is an important driver of the recruitment of intertidal invertebrates that reproduce through pelagic larvae by affecting larval transport and delivery. The intermittent upwelling hypothesis (IUH) posits that coastal pelagic larvae are driven offshore under intense upwelling or to depths under intense downwelling, while weak upwelling allows larvae to stay near the shore, thereby facilitating intertidal recruitment. The IUH thus predicts a unimodal relationship between Bakun’s upwelling index (BUI) and intertidal recruitment. The IUH has been supported by studies that plotted simultaneously single average values of recruitment and BUI for various coastal locations that collectively span downwelling to upwelling conditions. Based on its theoretical foundations, the IUH should also hold for a target location analyzed over the years provided enough interannual variation in BUI. On the Atlantic Canadian coast in Nova Scotia, upwelling varies interannually depending on wind patterns. Therefore, for a location that is representative of the abiotic and biotic conditions on this coast (Western Head), we tested the IUH by measuring annual intertidal barnacle recruitment and BUI for the pelagic larval season of barnacles for a period of 10 years (2014–2024). On this coast, BUI averaged for the barnacle larval season varied among years from mild downwelling to clear upwelling. Generalized additive modelling revealed a unimodal relationship between intertidal barnacle recruitment and BUI, thereby supporting the IUH. These results add this western ocean boundary to the known list of coastal systems where upwelling may influence intertidal invertebrate recruitment.

Introduction

In ecology, recruitment broadly refers to the appearance of juvenile organisms in a population (Haddon, 2001; Silvertown & Charlesworth, 2001). This process is key for the persistence of populations and may even influence community structure by altering the intensity of interspecific interactions (Menge & Sutherland, 1987; Connolly & Roughgarden, 1999). Studying the drivers of recruitment has therefore been a common theme in ecology. Due to fundamental differences across ecosystem types and organisms, studies have often focused on system-specific factors. For example, on marine shores that host abundant sessile invertebrates that reproduce through pelagic larvae, coastal oceanography is an important driver of benthic recruitment by affecting larval transport and delivery (Underwood & Keough, 2001; Gaines, 2007; Menge et al., 2019).

This kind of benthic–pelagic coupling has been mainly studied for rocky intertidal systems, which are those occurring on marine rocky shores between the highest and lowest tide marks. Studies were mainly done on temperate shores, where the most common sessile invertebrates with pelagic larvae are barnacles and mussels. Among other factors, the recruitment of these organisms is influenced by coastal upwelling and downwelling (Connolly, Menge & Roughgarden, 2001; Navarrete et al., 2005; Queiroga et al., 2007; Lagos, Castilla & Broitman, 2008). Coastal upwelling is the rise of deep waters to the surface of coastal environments as surface waters move offshore, while coastal downwelling involves the sinking of coastal water masses that are replaced by offshore surface waters moving onshore (Kämpf & Chapman, 2016; Jacox et al., 2018). On many coasts, these phenomena are caused by the combined action of winds and the Coriolis force. In the northern hemisphere, persistent alongshore winds blowing with the coast on the left (on the right in the southern hemisphere) generate an offshore surface Ekman transport that triggers coastal upwelling, while the opposite wind direction triggers coastal downwelling (Stewart, 2008). The intensity of these phenomena has often been measured with an index that is based on wind and geographic data (Bakun, 1973). Positive values of Bakun’s index denote upwelling, whereas negative values denote downwelling. Recently, it has been proposed that coastal pelagic larvae are driven offshore under intense upwelling or to depths under intense downwelling, while weak upwelling allows larvae to stay near the shore, thereby facilitating intertidal recruitment. The resulting intermittent upwelling hypothesis (IUH) thus predicts a unimodal relationship between Bakun’s upwelling index (BUI) and the recruitment of sessile intertidal invertebrates with pelagic larvae (Menge & Menge, 2013).

The IUH has been supported by some studies. For example, the recruitment of intertidal barnacles and mussels was unimodally related to BUI when plotting together data for several sites in Oregon, California, and New Zealand that collectively span a wide BUI range from downwelling to intense upwelling (Menge & Menge, 2013). Intertidal barnacle recruitment was also unimodally related to BUI when simultaneously plotting data from sites in Australia and South Africa that, in combination, span downwelling to upwelling conditions (Lathlean et al., 2019). Studies covering a narrower BUI range in Chile found that intertidal barnacle and mussel recruitment measured at the regional scale also decreased from weak to strong upwelling (Navarrete et al., 2005; Lagos, Castilla & Broitman, 2008). Another study spanning only low-to-high upwelling conditions found in Portugal also a negative relationship between barnacle recruitment and BUI (Fernandes et al., 2021). These results suggest that the IUH could be useful to predict benthic invertebrate recruitment in coastal systems, but wider testing is needed to avoid oversimplistic extrapolations as done for other ecological concepts in the past (Foster, 1990; Sagarin & Gaines, 2002). The present study tests the IUH for a western ocean boundary in Atlantic Canada.

We employed a dynamic sampling design that recognizes that both invertebrate recruitment (Kendall et al., 1985; Jonsson, Berntsson & Larsson, 2004; Navarrete, Broitman & Menge, 2008; Menge et al., 2011; Scrosati & Ellrich, 2016) and BUI (Kämpf & Chapman, 2016) vary interannually to some extent. On this basis, the IUH may be tested by analyzing the recruitment–BUI relationship for a location of interest surveyed over several years. While this approach may reveal a narrower BUI range than the joint consideration of various sites differing greatly in BUI (such as those studied by Menge & Menge (2013)), the IUH should theoretically hold for a target location provided enough interannual variation in BUI. On the Atlantic Canadian coast in Nova Scotia, upwelling can vary greatly between years in connection to changes in wind patterns (Scrosati & Ellrich, 2020). Therefore, this study tests the IUH using data on intertidal barnacle recruitment and BUI for a wave-exposed location on this coast surveyed over 10 years. We used wave-exposed habitats because they are often better than wave-sheltered habitats to uncover benthic–pelagic links with the ocean, as exposed habitats face open oceanic waters and thus are less influenced by land systems than semienclosed sheltered habitats (Tam & Scrosati, 2014).

Materials and Methods

We measured barnacle recruitment annually for 10 years (2014–2024) at Western Head (43.9896°N, 64.6607°W; Fig. 1). This is an extensive rocky headland that is representative of the abiotic conditions (Scrosati, Ellrich & Freeman, 2020) and intertidal communities (Scrosati et al., 2022) that characterize the southern Atlantic coast of Nova Scotia, Canada. Thus, it has been used before as a model location to study intertidal ecology in this region (Scrosati & Holt, 2021). The thermal properties of the intertidal habitats studied at Western Head (see below) have been described in a data paper based on half-hourly values of temperature measured by intertidal loggers for 5.5 years between 2014 and 2019 (Scrosati, Ellrich & Freeman, 2020).

Figure 1 Study location.

Left: map indicating the position of Western Head on the Atlantic Canadian coast in Nova Scotia. Right: wave-exposed rocky intertidal habitats at Western Head photographed at low tide in 2024. The photograph was taken by RA Scrosati.

On this coast, there is only one species of intertidal barnacle (Semibalanus balanoides). At Western Head, we measured barnacle recruitment at the mid-to-high intertidal zone in wave-exposed habitats (Fig. 2), where barnacles are typically most abundant. To determine the elevation of such places relative to chart datum (lowest normal tide in Canada), in 2014 we determined the upper intertidal boundary for wave-exposed habitats as the elevation of the sessile perennial species that occurred highest on the shore outside of tidepools and crevices (coincidentally, that species was S. balanoides). The upper intertidal boundary was 2.2 m above chart datum, similar to the maximum tidal amplitude for Western Head (Tide and Current Predictor, 2025). We then divided this vertical distance by three and located our targeted elevation (mid-to-high zone) just above the lower margin of the upper third of this vertical range. Thus, we measured barnacle recruitment at an elevation of 1.5 m above chart datum. Maximum water velocity (a proxy for wave exposure) measured at nearby wave-exposed intertidal locations can reach 12 m s−1 (Hunt & Scheibling, 2001).

Figure 2 Intertidal barnacles (Semibalanus balanoides) at Western Head.

Left: view at low tide of a wave-exposed intertidal habitat showing abundant barnacles at mid-to-high elevations facing open waters photographed in 2021. Right: barnacle recruits on the natural rocky substrate of one of the replicate quadrats (10 cm × 10 cm) cleared shortly before the recruitment season as seen at low tide on 3 July 2019. Both photographs were taken by RA Scrosati.

In Atlantic Canada, Semibalanus balanoides mates in autumn, broods in winter, and releases pelagic larvae in spring (Bousfield, 1954; Crisp, 1968; Bouchard & Aiken, 2012). The recruitment season (the period during which recruits appear on the substrate after the settlement and metamorphosis of the pelagic larvae) spans May and June on our coast, but most recruits appear in May. For example, frequent surveys done in 2019 at Western Head at the mid-to-high zone in wave-exposed habitats showed that recruits started to appear between 2–9 May and ceased to appear between 13–24 June, but more than 80% of the recruits recorded for that year were already on the substrate by the end of the third week of May (Scrosati & Holt, 2021). As of early July of that year, no recruit mortality had occurred (Scrosati & Holt, 2021). This temporal pattern also emerged based on observations done at Western Head and other intertidal locations on this coast since 2011, indicating that counts of recruit density done in late June or early July measure annual recruitment rates adequately. Therefore, we quantified annual barnacle recruitment as recruit density on the following 10 dates: 21 June 2014, 27 June 2015, 25 June 2016, 23 June 2017, 2 July 2018, 3 July 2019, 24 June 2021, 22 June 2022, 8 July 2023, and 3 July 2024 (data are unavailable for 2020 due to travel restrictions triggered by the COVID pandemic). To quantify barnacle recruitment, each year we cleared eight quadrats (10 cm × 10 cm = 1 dm2) on the substrate at the mid-to-high intertidal zone of wave-exposed habitats in late April (before the appearance of barnacle recruits). We removed all macroscopic algae and invertebrates from each quadrat using a metallic spatula and then we vigorously scrubbed the substrate with a wire pad. On our coast, these clearings made on the natural rocky subtrate yield higher recruitment rates than plates covered with Safety-Walk tape (Scrosati & Ellrich, 2018), a material used on other shores to measure barnacle recruitment (Menge, 2000; Lagos, Castilla & Broitman, 2008; Mazzuco et al., 2015). For each quadrat, we measured recruit density for the dates mentioned above using photographs taken at low tide above each quadrat (Fig. 2). Because of the wave-exposed nature of the studied habitats, taking quadrat photos could only be done safely during the lowest tides on days with the calmest possible seas. In 2015 and 2016, rough sea conditions allowed us to take clear photographs of only seven of the eight cleared quadrats.

We used BUI because, on the southeastern coast of Nova Scotia, upwelling is mainly caused by wind (Petrie, Topliss & Wright, 1987; Shan et al., 2016). For each year, we calculated BUI for the period during which upwelling and downwelling could in principle be most influential on the annual barnacle recruitment rate. The nauplius larvae of Semibalanus balanoides stay in the water column for 5–6 weeks (Bousfield, 1954; Drouin, Bourget & Tremblay, 2002), which is therefore when upwelling and downwelling could affect larval transport. Thus, as more than 80% of all recruits found in 2019 appeared between 2–9 May and the end of the third week of May (Scrosati & Holt, 2021), we deemed the period including April and the first three weeks of May as potentially the most consequential in terms of BUI for the annual barnacle recruitment rate. Therefore, for each year we calculated BUI for that period using wind data measured at Western Head Station (43.9900°N, 64.6642°W; Government of Canada, 2025), which is a land weather station located just 300 m away from the intertidal zone at Western Head. We calculated BUI in three steps (Scrosati & Ellrich, 2020). In the first step, we used equation 4.2 in Stewart (2008) to calculate wind stress (τ) as: τ=ρairCDU2

where ρair is air density (1.28 kg m−3), CD is the wind drag coefficient (0.0015, based on Kämpf & Chapman, 2016), and U is wind speed (in m s−1). Since wind speed data were available hourly, we calculated wind stress for each hour. In the second step, we used equation 2.2 in Kämpf & Chapman (2016) to calculate Ekman transport (M) as: M=τ/ρseawater|f|

where ρseawater is seawater density (1,026 kg m−3) and f is the Coriolis parameter. The Coriolis parameter depends only on latitude (φ) and was calculated as: f=4πsinφ/Tearth

where Tearth (86,400 s) is the period of the Earth’s rotation (Kämpf & Chapman, 2016). In the third and last step, we used equation 2.3 in Kämpf & Chapman (2016) to calculate BUI as: BUI=Mcosα

where the angle α is the difference between the average orientation of our coast (60°  measured clockwise from the north) and the angle (also measured clockwise from the north) denoting wind direction with the wind vector’s origin centered in the target location. For example, for our coast, a southwesterly wind blowing perfectly parallel to the shore yields α = 0°  and, thus, cos(α) = 1. Since wind direction data were also available hourly, we calculated BUI for each hour. To calculate mean BUI for the targeted period for each studied year, we averaged the corresponding hourly BUI values. An article by Jacox et al. (2018) provides a wider discussion on upwelling indices. For this study, we quantified BUI as cubic meters of seawater transported per second per 100 m of coastline (m3 s−1 (100 m of coastline)−1). For simplicity, the remainder of the text provides only the values of BUI without stating those measurement units explicitly.

We determined the relationship between BUI (predictor variable) and annual barnacle recruitment (response variable, measured as recruit density at the end of the recruitment season) by fitting a generalized additive model (GAM) to the data (Zuur et al., 2009). The GAM technique identifies the most suitable relationship between two variables without any pre-set function in mind. Due to the nature of the recruitment data, we calculated the GAM under a negative binomial distribution, which was supported by the absence of any evident structure in the corresponding predictor-vs-residuals plot. We calculated the percentage of variation in recruitment explained by BUI as the explained deviance (Zuur et al., 2009). We did these analyses with R version 4.2.3 (R Core Team, 2023), using the mgcv package to calculate the GAM. The underlying data on recruit density, winds, and BUI are freely available from the figshare online repository (Scrosati & Ellrich, 2025).

Results

The annual rate of barnacle recruitment (measured as recruit density at the end of the recruitment season) differed significantly among years (Kruskal–Wallis one-way analysis of variance, H 8 = 25.03; P = 0.003; Fig. 3). The highest annual average (80 recruits dm−2 in 2014) was more than 13 times higher than the lowest annual average (6 recruits dm−2 in 2016). The highest value of recruit density found for a single quadrat was 277 recruits dm−2 (recorded in 2021). For the 10 studied years, BUI averaged for the main barnacle larval season (April and the first three weeks of May; see Materials and Methods for rationale) ranged between −0.3 in 2024 and 21.2 in 2018 (Fig. 3).

Figure 3 Interannual changes in barnacle recruitment and BUI.

Left: barnacle recruit density (mean ± SE) measured at the end of the recruitment season of the studied 10 years. Right: BUI averaged for the pelagic larval season of barnacles for the studied 10 years.

The GAM describing the relationship between annual barnacle recruitment and BUI measured for the main barnacle larval season was significant (P < 0.001; Fig. 4). The explained deviance of the model was 22.4%. The recruitment–BUI relationship was unimodal, as recruitment peaked at intermediate values of BUI (7.4–11.4) and decreased towards higher BUI values denoting stronger upwelling and towards the negative BUI value that denoted downwelling (Fig. 4).

Figure 4 Barnacle recruitment versus BUI.

Generalized additive model (with a 95% confidence band) summarizing the relationship between annual barnacle recruitment and BUI averaged for the pelagic larval season of the studied 10 years. Each dot represents the value of barnacle recruit density measured for each quadrat cleared on the intertidal substrate shortly before the barnacle recruitment season.

Discussion

This study tested the IUH by examining interannual changes in intertidal barnacle recruitment as a function of BUI for a western ocean boundary in Atlantic Canada surveyed over 10 years. This dynamic sampling approach differs from recent studies that also tested the IUH, as such studies used single average values for different locations plotted simultaneously (Menge & Menge, 2013; Lathlean et al., 2019). The rationale for our approach was that, if enough interannual variation in BUI exists for a given shore, the IUH should also apply because of its theoretical foundations (see Introduction). For Western Head, BUI averaged for the period when most barnacle larvae are in the water (April and the first three weeks of May) showed interannual variation ranging from mild downwelling to clear upwelling conditions.

The GAM technique identified the most suitable recruitment–BUI relationship without having any preconceived function in mind. This modelling approach supported the IUH because barnacle recruitment peaked at intermediate BUI levels (7.4–11.4) for the range encountered at Western Head. Interestingly, the predicted decrease in recruitment with increasing BUI occurred at BUI values (between 18 and 21) that were lower than those denoting strong upwelling on other coasts, such as above 40 in eastern South Africa (another western ocean boundary; Lathlean et al., 2019) and above 150 in California (an eastern ocean boundary; Menge & Menge, 2013). Similarly, the predicted decrease in recruitment towards negative values of BUI (denoting downwelling) was evident at a value of BUI (−0.3) that was two orders of magnitude lower than values measured for eastern Australia and eastern New Zealand (between about −20 and −40), two other western ocean boundaries previously used to test the IUH (Menge & Menge, 2013; Lathlean et al., 2019). Overall, the results for Western Head support the notion that upwelling may influence the supply of larvae to intertidal habitats that ultimately affects intertidal recruitment (Menge & Menge, 2013). Quantifying the density of pelagic larvae in coastal waters in relation to changes in BUI at Western Head should help to further evaluate this notion.

Despite the significance of the GAM, the amount of variation in barnacle recruitment explained by BUI was moderate (22.4%). In part, this was a result of the natural variation in recruitment found among the surveyed quadrats each year, which was sometimes substantial. In intertidal habitats, biological variation at local scales can be large (Underwood & Chapman, 1998; Fraschetti, Terlizzi & Benedetti-Cecchi, 2005; Valdivia et al., 2011) and is often driven by small-scale changes in substrate properties, water motion, moisture at low tide, and light and thermal conditions (Whorff, Whorff & Sweet, 1995; Guichard, Bourget & Robert, 2001; Helmuth & Denny, 2003; Munroe, Noda & Ikeda, 2010; Ørberg et al., 2018; Catalán et al., 2023). Our modelling used recruitment data for each quadrat to preserve that variation. Therefore, precisely because of that high variation, it is remarkable that BUI was identified as an explanatory variable for barnacle recruitment.

Tests of the IUH that plotted together single values of recruitment for different locations obtained such values by averaging data for replicate sampling units, thereby minimizing influences of small-scale variation in recruitment within locations for the analyses. However, in those studies, a sizeable variation in recruitment was still not statistically explained by BUI (about 50% for barnacles and mussels in Menge & Menge 2013). Identifying the various sources of variation in intertidal invertebrate recruitment remains an area of active research. Current comprehensive views posit that upwelling and downwelling influence larval transport to the shore or away from it to a certain extent and that other factors play important roles once larvae are near the shore (Menge & Menge, 2019; Menge et al., 2019). Those factors include surface waves, internal waves, onshore breezes (Pfaff et al., 2011; Pfaff et al., 2015), and surf zone width (Shanks & Morgan, 2018; Shanks & Morgan, 2019; Shanks, 2019), for example. Surf zone width, however, may not explain interannual changes in recruitment at the same location, as that factor depends mainly on coastal geomorphology (Shanks & Morgan, 2019), which is a fixed property for any given location. Clearly, the potential nearshore drivers of recruitment that can change interannually must be evaluated together to understand the variation not explained by interannual changes in BUI at Western Head. All in all, the present study adds this western ocean boundary to the known list of coastal systems where upwelling may influence intertidal invertebrate recruitment.

We are grateful to Willy Petzold and Carmen Denfeld for field assistance and to Nicholas Jeffery and three anonymous reviewers for constructive comments on an earlier version of this manuscript.

Additional Information and Declarations

Competing Interests

Author Contributions

Data Availability

The authors declare there are no competing interests.

Ricardo A. Scrosati conceived and designed the experiments, performed the experiments, analyzed the data, prepared figures and/or tables, authored or reviewed drafts of the article, and approved the final draft.

Julius A. Ellrich performed the experiments, analyzed the data, prepared figures and/or tables, authored or reviewed drafts of the article, and approved the final draft.

The following information was supplied regarding data availability:

The data are available at figshare: Scrosati, Ricardo A.; Ellrich, Julius A. (2025). Data on barnacle recruitment, wind, and Bakun’s upwelling index for Western Head, Nova Scotia, Canada (2014–2024). figshare. Dataset. https://doi.org/10.6084/m9.figshare.28687250.v2.

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
