# Peer review of "Support for the intermittent upwelling hypothesis using 10 years of barnacle recruitment data from a western ocean boundary in Atlantic Canada"

_PeerJ, doi:10.7717/peerj.19470_

## Round 0.1 · original submission · Minor Revisions

· Academic Editor

Minor Revisions

Three reviewers provided comments on this manuscript, and I agree that the study is succinct if not perhaps a bit too short, and well-suited for PeerJ. I encourage the authors to address the minor comments from two reviewers on some small fixes and citations, and to only write "Semibalanus" in full the first time it is defined in the text. The authors should also add this Latin name to the Figure 2 caption.

Reviewer 2 had some additional concerns, and while I disagree with them that there is not enough data for the GAM and overall analysis, I do agree that some additional data could flesh out the results a little bit more. While not necessarily the point of this manuscript, a summary of the environmental conditions at this site (e.g., mean and range of air/water temperature if available, wind speed) which might be useful to local scientists. I am also curious if any other variables could affect barnacle recruitment year-to-year, such as storms (hurricanes) or microgradients in the high tide zone, for example in sunny versus more shade-covered areas (if any).

The results and conclusions however are well-supported by the data, and a GAM is appropriate for this analysis. Overall, I believe this manuscript requires minor revisions described by the reviewers, and perhaps some additional consideration on other factors that could influence barnacle recruitment on an inter-annual basis.

best,
Nick Jeffery

Reviewer 1 ·

Basic reporting

All aspects of basic reporting are met.

Experimental design

All aspects of experimental design are clearly presented and the analyses used valid and robust.

Validity of the findings

The study and its conclusions are relevant and timely. I have prepared a brief paragraph highlighting these strengths and providing a few points that the authors may want to clarify or discuss further (see Additional Comments). I expect this article to be a valuable contribution to ecology and oceanography.

Additional comments

The manuscript by Scrosati and Ellrich summarizes a ten-year monitoring of barnacle recruitment in a Nova Scotia coastal site and parallel estimates of upwelling intensity (BUI). This combined dataset allowed the authors to test an important hypothesis in ecology / oceanography, the intermittent upwelling hypothesis. The novel part of this study is the use of temporal (annual) rather than spatial data, which seems to be the norm for all the prior studies attempting the same goal. The manuscript is therefore relevant, not to mention that it is well written, with clear objectives and a methodology with all the relevant details. The results are brief and clear, and the Discussion addresses the most relevant literature on this subject. I have no doubts about the quality of the study, and I recommend publication after the authors consider a few points (indicated below). These are mainly clarifications or points where they may want to expand some statements for the benefit of the readers.

Some points to be considered.

Study area: Consider adding a couple of lines justifying the choice of location for this study.

Lines 77-78: A verb is missing in line 77. Perhaps: “… also FOUND a negative…”?

Lines 140-141. This is a bit of an odd statement. Are there other relevant instances where upwelling is not related to wind?

Line 188. The use of recruits per “dm2” may be common in rocky shore recruitment studies but should be defined/explained for those working on other types of habitats.

Line 188-189. Just for completion perhaps the two highest recruitment averages should be mentioned (2014 and 2021 are almost identical in the plot).

Lines 212-216. As the authors state, BUI upper values (18-21) are very low compared with those in other regions (>150 in some). This raises the question of whether higher BUI levels (in case they ever occur in the study area) would still be associated to low recruitment…? In other words, is there a chance that the BUI range recorded here was too narrow to describe a stronger association with recruitment? This applies to the other end of the range as well, as -03 is far from lower ranges between -20-40 measured in other studies. I’m not questioning the results, this is beyond the scope of the data collected, but it would be interesting to see if the authors decide to go further discussing this point.

Lines 228-230. Another statement that I find a bit odd. “Quantifying the variation… should help to further evaluate this notion…” That is what this study did, so maybe the idea was to suggest exploring further (longer-term scale) so wider variation in both BUI and recruitment levels allows for further testing of the IUH?

Line 239. Consider replacing the word “preserve” with “reflect” or “highlight”

Lines 241-242. Rephrase/clarify this statement. It appears as if you were stating that the comparison among locations had little to no small-scale variation to be worried about.

Lines 231-232. This is an important point: only 19% variation was explained by the model, and even though it outcome was significant, it is still far down from the 50% variation explained by other studies. The latter may be related to the fact that those spatial comparisons involved not only various locations in one region but also comparisons among distinct regions and/or continents. That highlights again a neat aspect of this study: using a temporal framework where BUI values were narrower, still offers (significant) support to this important hypothesis.

Reviewer 2 ·

Basic reporting

The objective is clear. The manuscript has been prepared professionally.
Results are relevant to the hypotheses.

Experimental design

Although data has been collected for 10 years, the number of replicated sampling units is too small (n=8). In addition, the explanation of why the data was collected only at one location isn't sound. To test a hypothesis across a large oceanographic realm, replicated locations are needed. It is needed to make sure that the sampling design can eliminate uncontrolled confounding factors that can interfere with the results when collection takes place at only one location. I believe that the authors might have a bigger dataset that can be used. I strongly recommend that the authors improve the manuscript by adding more data in the analyses.

Validity of the findings

As the results were obtained from a small dataset, the validity of the finding is debatable.

Additional comments

Line 77 add showed?
Lines 92-95 add references.
Line 98 How is the study site classified as a representative? Need more clarification.
Line 101 Why mid-to-high zone? Is the barnacle most abundant here?
Line 203-207 This should be in the introduction or method.
Line 207 What is "its theoretical foundation"?
Line 210-211 This should be in the method. I don't think the writing style is academic, please rewrite.
Line 232 The model explained 20% of variation because the dataset is too small.

Reviewer 3 ·

Basic reporting

This manuscript presents a test of the intermittent upwelling hypothesis, using 10 years of barnacle recruitment data at a single location instead of along a coastline, as previously done by others. Because the study is conducted at one location, it covers a narrower range of upwelling strengths, but the conclusions are similar to other studies in that the intermittent upwelling hypothesis holds: recruitment was highest at intermediate upwelling index values with the caveats that the upwelling index explains only 19% of the variation in recruitment and that recruitment was also high at low upwelling index values.
Although the analysis is simple and the findings are incremental, the work is well motivated, the writing is informative and clear, and both the data and analysis are appropriate. The figures are relevant, well labelled and described, and a reference is provided to the raw data, which I confirmed is available. Overall, the manuscript is a simple but elegant contribution with a sound methodology, which fits the objective of PeerJ.

I have only limited suggestions, but would like to see them addressed:

1) A detailed explanation of the Bakun index and other upwelling indices is beyond the scope of this manuscript, but Jacox et al. (2018) provide accessible explanations and context. Their work should be cited in the introduction and methods, which would be beneficial to the reader (e.g., their figure 3 shows that some studies use per m of coastline and others per 100 m of coastline).

Jacox et al. (2018). Coastal Upwelling Revisited: Ekman, Bakun, and Improved Upwelling Indices for the U.S. West Coast. https://doi.org/10.1029/2018JC014187

2) L. 241-242 – although I agree that using average recruitment values (provided that enough quadrats were used) can average over small-scale variation at a site; wouldn’t we expect variation in local properties or small-scale oceanographic processes (e.g., surface & internal wave conditions, presence/absence of retentive eddies) to be high between locations? The rest of the paragraph makes it clear that the authors understand this fact, but the first sentence should be clarified to align with that understanding.

3) L.92-95 – the statement is logical, but please provide citations and examples to support it.

4) L.207 – please expand on ‘theoretical foundations’ and clarify what is meant.

5) L. 106/113/143 – for consistency, choose S. balanoides for the remainder of the paper or the full name.

Experimental design

The research is within the scope of PeerJ; the research question is well defined, relevant and meaningful; the data collection methods is well established; and the analysis method is described in detail, allowing replication.

Validity of the findings

The results both confirm and expand prior findings; the underlying data are publicly available; conclusions are well stated.

Additional comments

No additional comments.

---

## Round 0.2 · Minor Revisions

· Academic Editor

Minor Revisions

We received two reviews of the revised manuscript, one of which accepted the manuscript and the other requests minor revisions. I do agree with Reviewer 2 that a couple of their comments were not adequately addressed in the first revision, and ask the authors to consider the three minor issues that will help clarify some topics in the paper, particularly the BUI. I ask the authors to consider these three suggestions through minor revisions, and thank the reviewers and authors for their useful comments and rebuttal.

Reviewer 1 ·

Basic reporting

Ok

Experimental design

Ok

Validity of the findings

Ok

Additional comments

The authors properly addressed all the questions raised. In my opinion, the revised version of the manuscript is suitable for publication.

Reviewer 3 ·

Basic reporting

Although my five initial comments were relatively minor, I had mentioned wanting to see them addressed. In the revised version of the manuscript, however, 3/5 comments were not addressed to my satisfaction; the revisions do not rise to the standard of the original manuscript (see below). [A] indicates the author's response, and [R] my response.

1) A detailed explanation of the Bakun index and other upwelling indices is beyond the scope of this manuscript, but Jacox et al. (2018) provide accessible explanations and context. Their work should be cited in the introduction and methods, which would be beneficial to the reader (e.g., their figure 3 shows that some studies use per m of coastline and others per 100 m of coastline).
Jacox et al. (2018). Coastal Upwelling Revisited: Ekman, Bakun, and Improved Upwelling Indices for the U.S. West Coast. https://doi.org/10.1029/2018JC014187

[A] Since the calculation of BUI involves various steps, we deem necessary to specify those steps in our manuscript so future researchers know how we obtained our values. Doing this is also important because the book by Kämpf and Chapman (2016) contains a typographical error in its formula for wind stress (RAS spotted that error and confirmed it with both J. Kämpf and P. Chapman), so we had to use the formula for wind stress stated in Stewart (2008). In other words, our manuscript provides all of the correct formulas needed to calculate BUI. By the way, we were aware of the article by Jacox et al. (2018). While it is not strictly necessary to cite it in our manuscript because we provide the original sources that we used to calculate BUI, we have nonetheless added its citation as a general reference on upwelling.

[R] I am not satisfied with how this suggestion was addressed. Jacox et al. (2018) provides a review of upwelling indices, which is certainly useful to readers who may be ecologists but not experts on upwelling. Simply adding Jacox et al. (2018) to an existing statement without explaining to the reader that the citation is to a recent review of upwelling indices, including Bakun, is a disservice. Please adjust the text to reflect this additional information, which is of use to a reader needing more information.

2) L. 241-242 – although I agree that using average recruitment values (provided that enough quadrats were used) can average over small-scale variation at a site; wouldn’t we expect variation in local properties or small-scale oceanographic processes (e.g., surface & internal wave conditions, presence/absence of retentive eddies) to be high between locations? The rest of the paragraph makes it clear that the authors understand this fact, but the first sentence should be clarified to align with that understanding.

[A] We have adjusted that sentence.

[R] The sentence now says: ‘Tests of the IUH that plotted together single average values for different locations were, by design, likely unaffected by small-scale variation in recruitment within the locations.’
However, the phrasing is not explicit, which makes it hard to interpret. Instead, please be explicit: ‘By averaging over several quadrats at each location, previous tests of the IUH minimized the effects of small-scale variation in recruitment within each location.”

4) L.207 – please expand on ‘theoretical foundations’ and clarify what is meant.

[A] This is a commonly used expression. The meaning of “theoretical foundation” is given precisely by those two words: The theory on which a given concept is founded. For the IUH, its theoretical foundation is explained in the Introduction.

[R] While I understand that the sentence is equivalent to saying ‘in theory, the IUH should also apply at a single site’, in the context of the first paragraph of the discussion, the current sentence makes for a poor justification of the study and is dissatisfying to the reader. There are several options to address this comment, including: 1) the sentence could end after ‘apply’ and remain vague but at least not allude to a more substantive explanation, or 2) it could be rephrased to imply that this is exactly what the study was testing. In its current state, a short summary of why it should still apply is warranted.

Experimental design

No comment.

Validity of the findings

No comment.

Additional comments

No comment.

---

## Round 0.3 · accepted · Accept

· Academic Editor

Accept

The authors have made their methods more explicit and clear, and added several citations that have improved the manuscript which I believe can now be accepted. I thank the two reviewers for their comments but note that I believe Reviewer 2 is justified in requesting some re-wording at line 207 re: the discussion on 'theoretical foundations', which was not an argument about its definition but rather rewording the sentence to note this is what the authors were testing. If the authors wish to use this suggestion, they can incorporate it before or at the proofing stage.
Best regards,
Nick